# Self-Guided Smartphone Application to Manage Chronic Musculoskeletal Pain: A Randomized, Controlled Pilot Trial

**DOI:** 10.3390/ijerph192214875

**Published:** 2022-11-11

**Authors:** Chao Hsing Yeh, Jennifer Kawi, Lauren Grant, Xinran Huang, Hulin Wu, Robin L. Hardwicke, Paul J. Christo

**Affiliations:** 1Cizik School of Nursing at UTHealth, The University of Texas Health Science Center at Houston, Houston, TX 77030, USA; 2School of Nursing, University of Nevada, Las Vegas, NV 89154, USA; 3School of Public Health, University of Texas Health Science Center at Houston, Houston, TX 77030, USA; 4McGovern School of Medicine, The University of Texas Health Science Center, Houston, TX 77030, USA; 5School of Medicine, Johns Hopkins University, Baltimore, MD 21205, USA

**Keywords:** auricular point acupressure, chronic musculoskeletal pain, smartphone application, self-management

## Abstract

Objective: The goal of this study is to evaluate the feasibility and efficacy of an auricular point acupressure smartphone app (mAPA) to self-manage chronic musculoskeletal pain. Methods: A prospective, longitudinal, randomized, controlled pilot trial was conducted using a three-group design (self-guided mAPA (*n* = 14); in-person mAPA (*n* = 12); and control (*n* = 11)). The primary outcomes included physical function and pain intensity. Results: After a 4-week APA intervention, participants in the in-person mAPA group had improved physical function of 32% immediately post-intervention and 29% at the 1M follow-up. Participants in the self-guided mAPA group had higher improvement (42% at post-intervention and 48% at the 1M follow-up). Both mAPA groups had similar degrees of pain intensity relief at post-intervention (45% for in-person and 48% for the self-guided group) and the 1M follow-up (42% for in-person and 45% for the self-guided group). Over 50% of the participants in each group reached at least 30% reduced pain intensity at post-intervention, and this was sustained in the mAPA groups at the 1M follow-up. Approximately 80% of the participants in both mAPA groups were satisfied with the treatment outcomes and adhered to the suggested APA practice; however, participants in the self-guided group had higher duration and more frequency in APA use. The attrition rate was 16% at the 1M follow-up. No adverse effects of APA were reported, and participants found APA to be beneficial and the app to be valuable. Conclusions: The study findings indicate that participants effectively learned APA using a smartphone app, whether they were self-guided or received in-person training. They were able to self-administer APA to successfully manage their pain. Participants found APA to be valuable in their pain self-management and expressed satisfaction with the intervention using the app.

## 1. Introduction

Chronic musculoskeletal pain (CMP) is a major public health concern as the most common self-reported pain condition, and it affects about 126 million adults in the United States [1,2,3,4]. It is the most common complaint in chronic pain, occurring particularly in the joints and back [5] and lasting beyond 3–6 months (typical healing time) [6], with expenses over USD 635 billion annually for costs related to medical care and disability, as well as lost wages [6,7]. Despite a range of therapeutic approaches, chronic musculoskeletal pain (CMP) continues to be a major health concern [1,2,3,4,8,9,10,11,12,13].

Clinical guidelines advocate for nonpharmacologic therapies and self-management (SM) of chronic pain [5,14], but providers and patients alike are met with challenges toward nonpharmacologic and SM options for pain. Opioids and analgesics are the common treatments for chronic pain in current US healthcare system. However, pharmacotherapy is associated with numerous adverse side effects and has contributed to the deadly opioid crisis [15]. On the other hand, existing nonpharmacologic therapies have barriers to their implementation [16,17]. Variations in preferred treatment for chronic pain suggest clinical equipoise among healthcare providers [18,19]. Exercise has moderately improved CMP in randomized, controlled trials (RCTs) [14,20], but there is lack of long-term adherence and sustained effect [21]. Even a simple walking program does not have a sustained effect over time [22]. Multidisciplinary rehabilitation programs are recommended, but they are costly and time consuming. Acupuncture was recently included in the current guidelines for back pain and, as of 2020, covered by Medicare [23]. However, even with these advances, widespread implementation is limited by frequent office visits and a lack of access to licensed acupuncturists [21,22,24,25,26,27]. Other treatments (mindfulness-based stress reduction [28,29,30], tai chi [31,32], yoga [33,34], cognitive behavioral therapy [30,35], and spinal manipulation [36]) have challenges that minimize scaling up, including access, insurance coverage, and patient and healthcare provider buy-in. Further, although SM is strongly recommended in the Institute of Medicine’s National Pain Strategy and plays a central role in controlling pain and maximizing function, especially since a cure for CMP is currently unrealistic [7], evidence suggests that SM alone does not result to clinically sustainable and significant effects in pain or physical function [37,38,39,40]. 

Similar to acupuncture, auricular point acupressure (APA) applies a stimulus to ear acupoints without the need for needles and is a promising intervention for the self-management of chronic pain. APA is viewed as a micro-acupuncture technique from traditional Chinese medicine that uses ear acupoints to treat pain or symptoms [41,42,43]. When a body symptom or pathology appears, it is projected to the ear based on somatic reflexology [41,42,43]. Ear points become active, which can be detected with an electrical point finder due to low skin resistance or pressure allodynia while probing [44]. Once the ear points are identified, stimulation can be performed with acupuncture needles, pellets or seeds, or electric stimulation [45,46] to achieve a therapeutic effect. In APA, small pellets (e.g., Vaccaria (nonmedicinal) plant seeds) are used, taped securely onto specified ear points based on the location of body pain, and pressed by the patient intermittently throughout the day at any time or place to manage pain [44,46]. The underlying mechanism for APA on pain relief is still not well-understood. In auricular acupuncture, or APA, pain related to nerves that are irritated can be relieved by the normalization of reflex pathways that are pathological and hypersensitive; these pathways interconnect the somatotopic brain and the ear microsystem [41,42,43]. When ear reflex points connecting to the brain’s somatotopic reflex system are stimulated, an abnormal brain pattern can correct its pathological reflex centers [47,48], change levels of anti- and pro-inflammatory biomarkers [49,50], and induce reflex reactions to relieve body pathology [41,45,46]. Correlations of ear points and brain pathways have been supported by functional magnetic resonance imaging (fMRI) [47,48]. 

In order to scale up APA, we initially developed a smartphone app (mAPA) [51,52]. This app originally included two videos (~5 min educational and ~3 min demonstration videos). We previously tested this initial app on 30 participants with CMP; they reported averages of 30% pain relief, 35% improved pain interference, and 40% improved function after 4 weeks of self-guided and self-administered APA. During post-intervention interviews, some participants said they had to view the videos repeatedly to identify ear point locations accurately; some were also unable to find ear tenderness points using the probe provided by the study. Five patients indicated that in-person training would be helpful or that they needed reminders to stimulate ear points. Based on these items of qualitative feedback from prior study participants [51,52], the purpose of this current study is to further refine our mAPA content (videos) to maximize the beneficial effects of our APA app on CMP. We subsequently test this app in two mAPA groups (self-guided and in-person) and a control group (waitlist, education-enhanced control) in order to examine the preliminary impact of our intervention on CMP outcomes. We decide to include a waitlist group (control) to allow everyone the opportunity to access our intervention; we also have two mAPA intervention groups (self-guided and in-person) to evaluate any differences in outcomes between the two groups. Then, we aim to evaluate participant experiences of APA and our revised app. The overall questions are, “how were your experiences with APA” and “how were your experiences with the smartphone app”.

## 2. Materials and Methods

This was a prospective, longitudinal, pilot waitlist RCT. Participants were randomly assigned to: (1) self-guided mAPA, (2) in-person training with mAPA, or (3) waitlisted education-enhanced control groups. The self-guided mAPA group was instructed virtually on APA protocol in an approximately 15 min session, while the in-person group received in-person APA training at the recruitment site for a similar timeframe. In addition to using the same timeframe, we provided chronic pain education to our control group after recruitment and advised this group that they were waitlisted. Participants in the control group were re-randomized into one of the mAPA groups once they completed their 1M follow-up. 

Data from all groups were collected at pre- (T1, or first time point) and post-intervention (T2, or second time point), as well as at a 1 month (1M) follow-up appointment (T3, or third time point). Participants were instructed to install the APA smartphone app on their phones and were advised on how to use the app to self-administer APA for 4 weeks. At T2, immediately after the four-week intervention, participants were asked to complete questionnaires similar to the surveys from the baseline visit; an interview was conducted asking about their experiences with the smartphone app and APA. At T3 (1 month after completion of the intervention), the participants completed another follow-up assessment. 

APA treatment protocol. Detailed information about our APA treatment protocol has been detailed in our other publications [51,52]. The protocol included the following: (1) locating ear points that corresponded to the painful body part, (2) securing seeds with waterproof tape on the identified ear points, and (3) pressing seeds to achieve pain relief [44,53]. Participants were advised to evenly press the secured seeds over each ear point without rubbing in order to avoid skin damage or seed movement from the specific ear point for at least three minutes three times per day (nine minutes total/day), even if they were not having pain at that moment. Optimal pressure on the ear point was achieved when the participant felt mild discomfort or localized tingling. The treatment duration was four weekly cycles overall. Each cycle included five days of wearing seeds and two days with no seeds. Participants removed the taped seeds at the end of the fifth day, allowed the ear points to rest for two days, and re-applied after 2 days. This cycle reduced the risk of allergic reaction to the waterproof tape and allowed the ear points to rest and restore sensitivity before the next treatment cycle. After four weekly cycles of APA treatment, participants were interviewed about their experiences with mAPA and the APA intervention. In this study, the mAPA (Figure 1) was refined as indicated below based on feedback from participants in our previous study [51]. 

(1) Shorter videos were developed so that participants can choose specific content based on a specific need. The content of the videos included: a. overview of theoretical background of APA (video length: 3:29), using APA for pain (0:50), locating ear points (1:03), and stimulating ear points (1:21); b. overview of master ear points (3:00) and specifics on master points, such as the Shenmen point (0:41), subcortex nervous point (0:48), and ear center point (0:45), which all serve to enhance the relief of pain and pain-related symptoms [46,54]; and c. points related to body pain, such as lower back (1:49), neck (0:54), shoulder (0:56), hand (2:04), hip (0:40), knee (1:24), and foot (2:12) pain. 

(2) Diagrams of ear points related to the painful body part, including the spine, upper limbs, and lower limbs, were included [45,46].

(3) A question and answer section (commonly asked questions with answers) and an APA instruction sheet (providing succinct reminders related to APA protocol) were included.

(4) Using a theory-driven foundation (Bandura’s self-efficacy theory), an individualized dashboard for self-monitoring, as well as data-driven motivational messages, were included to promote APA practice, retention, and sustainability. Bandura’s self-efficacy theory [55,56] refers to an individual’s belief or confidence in one’s ability to perform the actions necessary to reach a goal; it is a significant precondition to self-management [55]. One develops self-efficacy via 4 major sources of information: personal accomplishment, vicarious experience, verbal persuasion, and emotional arousal [55]. These 4 sources were targeted to facilitate self-management, and we used motivational messages based on a participant’s APA practice adherence score. The individualized dashboard showed graphs of participants’ pain outcomes and frequencies of APA use based on surveys they completed during the study. 

### 2.1. Participants and Study Setting

Eligibility was determined based on the following criteria: (1) 18 years or older, (2) able to read and write English, (3) CMP for at least 3 months of duration, (4) average pain intensity of ≥4 on an 11-point numerical scale for the past week, (5) smartphone user, and (6) able to apply pressure to taped seeds on ears. Participants were excluded from the study if they had any latex allergies (due to tape used to secure seeds on ear points). This was a multisite study conducted at universities and healthcare settings located on the east and west coasts of the United States. 

### 2.2. Study Instruments

Roland–Morris Disability Questionnaire (RMDQ) [57]. The RMDQ is a 24-item measure used to assess the impact of pain on daily function and abilities. Participants were asked to answer yes or no on statements related to physical function. Scores ranged from 0 to 24 (no disability to maximum disability). RMDQ is a valid, reliable, and sensitive measure, with substantial construct validity [57,58]. A 30% improvement (reduction) from baseline is considered a clinically meaningful change on the RMDQ [59]. 

Pain Intensity (Numeric Pain-Rating (NRS) Scale). Pain intensity (average pain) was measured by self-reported NRS (range of 0–10, with higher score indicating more pain). A reduction of 30% in pain intensity defines pain relief, thereby evaluating the magnitude of change in pain intensity. This is considered a “moderate clinically important difference” by the Initiative on Methods, Measurement, and Pain Assessment in Clinical Trials (IMMPACT) [60] and is used to define a significant effect of an intervention. 

Pain-Catastrophizing Scale (PCS) [61]. The PCS is a 13-item self-report measure evaluating negative and exaggerated interpretations of pain. Participants were asked to reflect on painful experiences in the past and indicate the level to which they experienced each of the following subscales when feeling pain: rumination (4 items), magnification (3 items), or helplessness (6 items). The PCS is a 0–4 Likert scale with scores ranging from 0 to 52 (“not at all” to “all the time”). A higher PCS score indicates stronger pain catastrophizing. The PCS was found to be moderately correlated with other measures (e.g., depression, fear, anxiety, and negative affect), but only the PCS was significantly predictive of pain intensity [61]. 

Fear Avoidance Beliefs Questionnaire (FABQ) [62]. The FABQ [62] focuses on patients’ beliefs of how their pain is affected by physical activity (4 items) and how work affects their pain (7 items) [62]. The FABQ has a 0–6 Likert scale ranging from 0 (completely disagree) to 6 (completely agree). This instrument was developed and its content validated based on fear and avoidance behavior theories, particularly in relation to physical activity and pain [62] 

PROMIS 29 [63]. The subscales of “physical function” (4 items), “anxiety” (4 items), “depression” (4 items), and “fatigue” (4 items) of PROMIS 29 V2.0 were used. Each subscale’s scores range from 4 to 20 (higher scores indicate more symptoms or functions measured). PROMIS 29 has established validity and reliability [64,65], and it is widely used in the United States.

Patient Global Impression of Change (PGIC) [66]. The PGIC was used to measure overall impression of change (improvement) after completion of the study (smartphone app featuring APA). PGIC scores reflect a patient’s belief about treatment efficacy. It is a 7-point scale of overall improvement rating ranging from “no change” (1) to “very much improved”. The PGIC has established psychometric properties identifying clinically significant changes in subjective outcome measures [66]. 

Treatment Satisfaction [67]. This is a one-item scale that was used to assess the participants’ satisfaction with APA treatment. Scores were either 1 (“completely satisfied”), 2 (“somewhat satisfied”), or 3 (“not satisfied”). This survey was used effectively in our previous APA study on low back pain [67]. 

APA Practice [67]. This is a two-item, open-ended scale collecting data on how participants practiced their APA. Specifically, the scale asks “how many times did you press the seeds per day” (frequency) and “how long did you press the seeds each time” (duration). APA adherence to practice was defined by the number of participants who followed at least two-thirds of the suggested pressing time (at least two times per day or two minutes each time) [67,68,69]. This measure has been used successfully in our previous studies on low back pain [67,68,69]. 

Demographic and Health History Data. A survey was used to collect demographic information (e.g., age, marital status, educational level, employment, and estimated income), pain location, and pain medications used from the participants. 

### 2.3. Procedure

This study was approved by the Institutional Review Board (IRB) at an east-coast university using a single IRB mechanism with ceding from a west-coast institution. The study was conducted from October 2021 to June 2022. We advertised information regarding the study and recruited using online flyers distributed at our study settings. We obtained informed consent and collected baseline data. Participants installed the revised APA smartphone app on their phones and received their APA kits (included a probe to identify ear points and seeds secured with waterproof tape). They proceeded to self-administer APA for four weeks to self-manage their pain using the mAPA as a self-management tool to practice APA. All APA sessions were conducted by the first author (Yeh). Study outcomes were assessed at baseline (T1), weekly (W) during the intervention (W2, W3, and W4), immediately post-intervention (T2), and at a 1-month follow-up post-intervention (T3). Data collection was through Research Electronic Data Capture (REDCap), a secure data management system that is IRB-approved and HIPAA-compliant. A brief interview was conducted at T2 exploring participants’ experiences with APA and the mAPA. The interviews were recorded so that transcripts could be reviewed for analysis. Participants received gift cards of USD 20 for each study visit (up to USD 60). 

### 2.4. Data Analysis

The intent-to-treat approach, which includes all participants enrolled—regardless of treatment received, adherence, or withdrawal—was used for analysis. Missing values for any outcome variable were replaced with “last value carried forward.” Descriptive statistics were used for presenting demographic characteristics and results of the study measures. Parametric descriptive analyses (means and standard deviations) were used to examine the outcomes. The analysis sample was used as the denominator for all percentages, unless otherwise specified. The subscales in PROMIS 29 V1.0 were summarized using T-scores, which were generated by HealthMeasures (www.healthmeaures.net accessed on 22 October 2022) using PROMIS Wave 1 as the calibration sample. The adherence rate was determined based on the number of participants who were able to perform APA for at least two-thirds of the suggested pressing time (at least 2 times/day or 2 min/time) in order to ascertain the feasibility of practicing APA at home. A cut-off point of 30% improvement for primary outcomes (pain intensity and physical function) at the primary endpoint (1M) was used [70]. No formal hypothesis testing was performed due to the small sample size. Effect size as standardized differences in the mean between two means (Cohen’s d; mAPA groups compared to control in the mean change from baseline) was calculated; this was helpful to inform sample sizes for future RCTs. Data analyses were performed using SAS 9.4 and R 4.2.0. For the interview data, audio-recording transcripts were analyzed through conventional qualitative content analysis [71]. Common experiences, including recommendations, were extracted to understand participant experiences of APA and the smartphone app.

## 3. Results

### 3.1. Recruitment

Figure 2 demonstrates the flow of participant recruitment. A total of sixty-seven participants contacted the study coordinators of the site between September 2021 and November 2021 indicating that they were interested in the study. Thirty participants were excluded because they either did not meet the study inclusion criteria (*n* = 11) or did not respond when contacted after they expressed initial interest (19). Consequently, 37 participants were randomized into self-guided mAPA (*n* = 12), in-person mAPA (*n* = 14), and control (*n* = 11) groups using a computer-generated randomization tool. Overall, the attrition rate was low. At the 1M follow-up visit, six participants were lost to follow-up with a lack of response to contact (attrition rate = 16%).

### 3.2. Demographic Characteristics of Study Participants

Table 1 presents the demographic characteristics of the participants. The average ages of the participants in the three study groups were similar at about 50 years. The majority were female (*n* = 27, 73%). Many were white (*n* = 21, 57%), 43% (*n* = 16) were currently employed, and 95% (*n* = 35) had some form of college degree or higher. For duration of CMP, the majority of the participants reported pain for at least 1 year or longer (*n* = 35, 95%). There were 92% of the participants who reported more than one pain location in their bodies. The most common pain location was back (*n* = 18, 49%), followed by neck (*n* = 6, 16%), foot (*n* = 4, 11%), knee (*n* = 3, 8%), and hip (*n* = 3, 8%). All study participants were taking some type of medication(s) for their pain (Table 2), with 6 (16%) participants in the mAPA groups indicating no use of pain medication at the 1M follow-up assessment. 

### 3.3. APA Treatment Outcomes

Table 3 presents the study outcomes for each study group. 

#### 3.3.1. Physical Function (RMDQ)

Participants in the in-person mAPA group achieved 32% improvement in physical function post-intervention and 29% improvement at the 1M follow-up assessment compared to baseline levels. Among the participants in the in-person group, at least four (29%) achieved a minimum of 30% clinically significant improvement in physical function at the post-intervention and at 1M follow-up time points. Participants in the self-guided mAPA group had 42% improvement of physical function post-intervention and 48% improvement at the 1M follow-up visit, respectively. At least 25% of self-guided mAPA participants (*n* = 3) achieved a minimum of 30% clinically significant improvement of physical function at the post-intervention and 1M follow-up time points. Participants in the control group had 6% improvement post-intervention and 25% improvement at the 1M follow-up assessment, with no one reaching the minimum of 30% clinically significant improvement. 

#### 3.3.2. Pain Intensity (Average)

Participants in the in-person mAPA group experienced reduced (45%) pain intensity post-intervention, with 50% of the participants (*n* = 7) achieving at least 30% clinically significant improvement in pain intensity. The in-person mAPA group also had 42% reduced pain intensity at the 1M follow-up assessment, with 57% of the participants (*n* = 8) achieving at least 30% clinically significant improvement in pain intensity. Self-guided mAPA participants also reported a similar average pain reduction (48% post-intervention and 45% at the 1M follow-up visit) with 50% (*n* = 6) experiencing at least 30% clinically significant improvement in pain intensity post-intervention and 42% (*n* = 5) at the 1M follow-up assessment. Changes for the study participants in the control group were minimal (ranging from 1% to 11% post-APA and at the 1M follow-up visit), with no one achieving at least a 30% reduction in pain intensity.

#### 3.3.3. Pain Catastrophizing and Fear Avoidance

Participants in the in-person mAPA group had a higher reduction in pain catastrophizing (47% post-intervention and 52% at 1M follow-up assessment) compared to participants in the self-guided group. The fear avoidance of physical activity among participants in the in-person training group had 30% improvement post-intervention and 22% at the 1M follow-up visit, while participants in the self-guided mAPA group only had 2% improvement post-intervention but 51% improvement at the 1M follow-up visit. 

#### 3.3.4. Anxiety, Depression, Sleep Quality, and Fatigue (PROMIS 29)

Table 3 shows that changes in the scores at both data timepoints (T2 and 1M) were modest compared to baseline levels among the three study groups. 

#### 3.3.5. Patient’s Global Impression of Change

As noted in Table 3, participants indicated in both mAPA groups that they gradually experienced mild-to-moderate overall improvement during the intervention. No improvement was noted in the control group.

#### 3.3.6. Satisfaction with APA

In Table 4, the majority of the participants reported some or complete satisfaction with APA. There were more participants in the self-guided mAPA group who reported being completely satisfied compared to the in-person mAPA group. However, one participant in the self-guided mAPA group was not satisfied at either 4 weeks or 1 month after the 4-week intervention. Overall, the majority (~90%) of the participants were satisfied with the intervention in both mAPA groups. 

### 3.4. Adherence to APA Practice

All participants in both mAPA groups adhered to the suggested frequency (at least two times per day) and duration (at least two minutes each time) for a total of six minutes per day minimum in pressing the seeds. Participants in the in-person mAPA group practiced APA at least three times per day post-intervention, which decreased to 2.1 at the 1-month follow-up assessment. Participants in the self-guided mAPA group had a slightly higher overall average frequency in pressing their seeds (3.8 vs. 2.9) and a higher overall average duration in pressing their seeds each time (3.5 vs. 2.9). These all met the minimum suggested duration and frequency for APA use. 

### 3.5. Qualitative Findings

No remarkable differences were noted among responses from the self-guided vs. in-person participants at the recruitment sites. There were three overall themes noted from the interview responses *(N* = 28) addressing participant experiences of APA and the app. First, participants generally found that APA was beneficial for their pain and pain-related symptoms. Common words used by the participants related to APA were “like”, “simple and easy”, “accessible”, and “helped with pain symptoms” (for example, “my pain was lessened a lot”, “can sleep better and more comfortable”, and “helped in different areas of pain in my body”). They denied any adverse effects from their APA use. Some verbalized decreasing use of other pain-relieving measures and the ability to do more physical activities: “using less CBD”, “did not have to take Ibuprofen during treatment”, and “walking and dancing are more tolerable”.

The second theme was that the participants felt the app was valuable and “full of useful information”. Common words indicated by the participants were “helpful, “useful”, and “satisfied”. For example, one participant said, “I liked the videos especially in replacing the seeds; I was able to see the location of the ear points with the app if [I was] doing the right thing”. Others stated, “I was able to watch [the videos] several times especially to help with [ear point] placement” and that “the videos were nice and short which was good, they were descriptive enough”.

The last theme was that participants recommended improvements that could be helpful moving forward. We found that the motivational messages were “a nice touch” and “inspirational”, according to the participants, to facilitate adherence and retention, but some noted that these could be enhanced further to “include other recommendations such as stretching.” The study procedures were found to be thorough and acceptable, and the individualized dashboard for the self-monitoring of data was helpful and “made me accountable as a reminder to practice APA” and “incorporate [APA] into daily routine.” Some participants recommended improving inclusivity (various skin tones in ears shown in app videos), having a Spanish-speaking app or translation, and allowing for app internet access, as well as accessibility on other platforms (e.g., iPad, laptop, or desktop) apart from a smartphone.

## 4. Discussion

Based on feedback from our previous mAPA study [51], we further developed, revised, and pilot-tested our refined mAPA in this study using a theory-driven (Bandura’s self-efficacy) foundation and motivational messages to promote adherence and sustainability in APA practice. Although efficacy testing is not a goal for pilot studies [72], we found that 4 weeks of APA could improve physical function by 32% post-intervention and by 29% at the 1M follow-up assessment for in-person mAPA, while participants in the self-guided mAPA group had higher improvement (42% post-intervention and 48% at 1M follow-up visit). Both mAPA groups had similar degrees of pain intensity relief at the post-intervention timepoint (45% for in-person and 48% for self-guided group), as well as at the 1M follow-up visit (42% for in-person and 45% for self-guided group). Based on these findings, we found that mAPA could be used effectively as a self-guided tool; participants in both mAPA groups thought that the videos and content in the app were well-produced. Participants in the self-guided mAPA group had the same degree or even slightly better results on some study outcomes (i.e., physical function and pain intensity) and better adherence to APA practice compared to the in-person mAPA group. This is important because it indicates that APA could be self-learned through a smartphone app to effectively manage pain and pain-related symptoms without the necessity for in-person training. This has significant implications toward improving and increasing access to APA. At times when in-person visits are limited or when access to face-to-face care is challenging, we found that self-guided APA training was possible and could be successful. 

It is very important to compare our results to those of other nonpharmacological pain therapies recommended in clinical practice by the American College of Physicians [73]. For instance, the magnitudes of improvement in pain (2.4, NRS, 0 to 10) and function (−1.4 in RMDQ) at the 1M follow-up assessment for our in-person mAPA group exceeded the moderate strength of evidence documented for exercise (−0.93 in pain intensity using a visual analogue scale (NRS, 0 to 10) short-term and −0.47 long term), multidisciplinary rehabilitation (−0.55 in NRS short-term and −0.27 long-term), mindfulness-based stress reduction (−0.64 in the NRS and −1.37 in the RMDQ), acupuncture (−0.72 in the NRS), and cognitive behavior therapy (−0.60 in the NRS short-term) [73]. For pharmacologic interventions such as nonsteroidal anti-inflammatory drugs (NSAIDs), which are routinely used in clinical practice, NSAIDs only resulted in a mean difference in pain intensity compared to baseline level of −3.30 (95% CI, −5.33 to −1.27) on a 0 to 100 NRS with a median follow-up period of 56 days and a mean difference of −0.85 (95% CI, −1.30 to −0.40) from the RMDQ at 85 days of follow-up when compared to the placebo group [74]. Further, compared to acupuncture (both invasive (use of needle) and passive (administered by a licensed practitioner) and needing at least eight treatment sessions), APA is a relatively simple and an active approach (patients need to stimulate their ear points) with the capability of achieving a moderate magnitude effect at 1M of follow-up. Most other treatments, including acupuncture, need to be repeated multiple times with healthcare provider reliance to produce and sustain an effect, contributing to high healthcare costs. 

Another significant finding was that we were able to retain our control group using a pain-education-enhanced waitlist approach. This allowed us to sustain their participation in the study without significant attrition while providing them the opportunity to access our intervention. We also found that the participants were satisfied with the short videos and the specific content (e.g., specific ear point locations based on body part in pain) for ease of use, navigation, and improved learning outcomes. The demonstration videos emphasized using the probe to assist in identifying ear points for treatment. However, the current version of mAPA used was only in English and was not inclusive of other ear skin tones, as reflected in the feedback. In future refinements, the app videos can include other ear skin tones (e.g., White, Black, and Asian) and other languages (i.e., Spanish) to facilitate access. Additionally, an internet-enabled mAPA can allow patients to view APA content on a computer, tablet, or smartphone at any time or place based on their preferences. 

A data-driven, individualized dashboard was added in our current mAPA to display visual summaries of study outcomes (e.g., pain intensity, interference of activity, and APA practice duration and frequency). A motivational message based on participant APA practice, popped up in the screen to facilitate self-monitoring and promote adherence. Based on the study findings, participants in both mAPA groups had higher APA practice levels but decreased slightly at the 1M follow-up visit. Thus, continued promotion of APA practice and sustainability for long-term treatment effects are important. 

Considering the recommendations for use of pilot studies as a critical stage in the development and testing of interventions [72], we found that we were able to recruit and randomize our target population within 3 months and retain our participants at 16% attrition, with strategies needed for long-term sustainability. APA practice adherence rates met the minimum recommended duration and frequency of APA use while, at the same time, having a favorable impact on pain and pain-related symptoms based on our quantitative and qualitative data.

Although we obtained promising results, the interpretation of the current study findings is limited by the unblinding of the first author, who was the treating clinician, as well as by the small sample size due to the nature of the pilot study and the short-term follow-up period (1 month). Future studies need to address these shortcomings. 

## 5. Conclusions

In conclusion, we found that our revised mAPA was an ideal self-guided tool for individuals to learn and self-administer APA to manage their CMP. Larger studies are needed to determine the efficacy of mAPA on CMP. 

## Figures and Tables

**Figure 1 ijerph-19-14875-f001:**
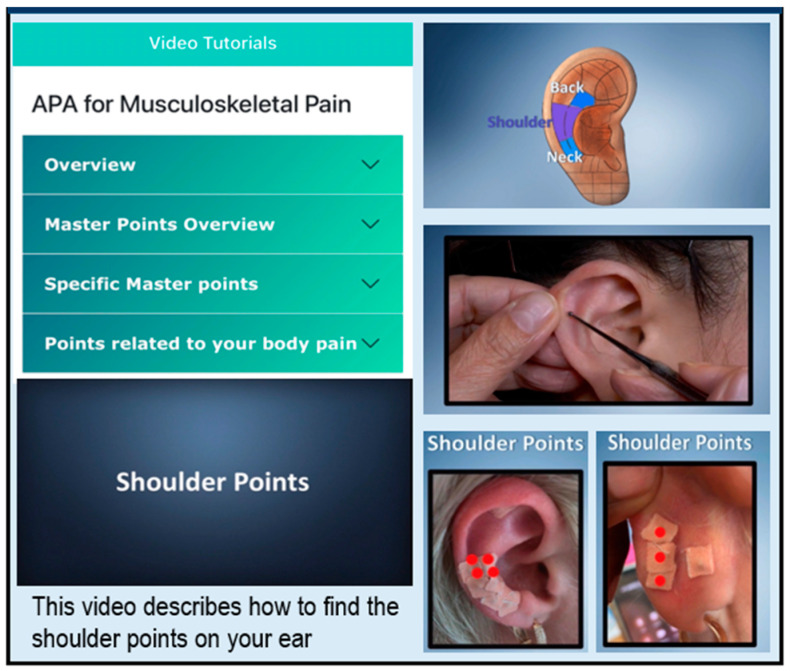
App for chronic musculoskeletal pain.

**Figure 2 ijerph-19-14875-f002:**
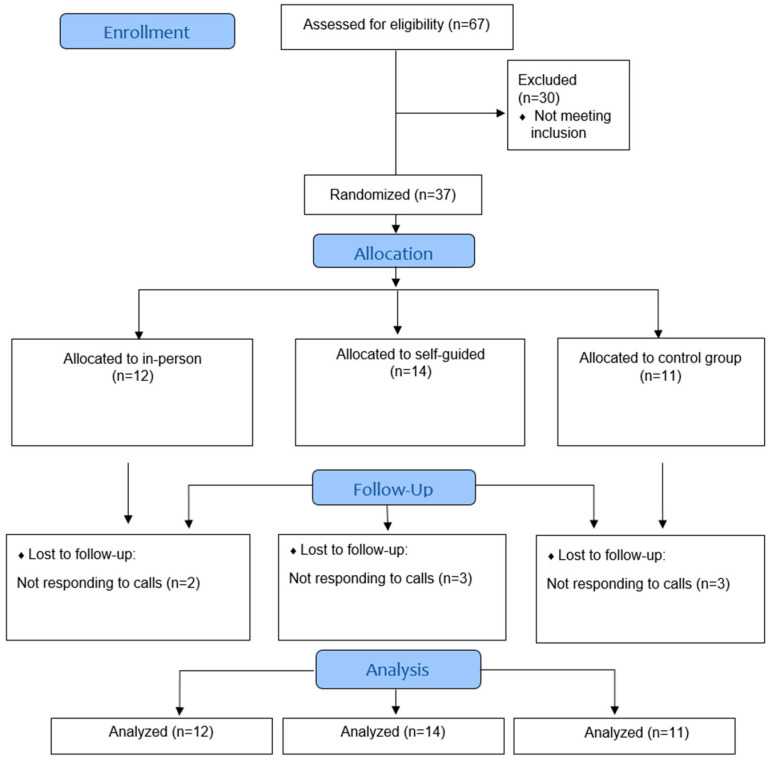
Enrollment flow diagram.

**Table 1 ijerph-19-14875-t001:** Characteristics of the participants (*n* = 37).

	In-Person mAPA (*n* = 14)	Self-Guided mAPA (*n* = 12)	Control (*n* = 11)	Overall (*n* = 37)
Age, years				
Mean (SD) (range)	50.4 (20.70) (21–81)	51.2 (17.49) (19–75)	50.1 (13.95) (20–70)	50.5 (17.39) (19–81)
Gender, *n* (%)				
Female	11 (79)	8 (67)	8 (73)	27 (73)
Male	3 (21)	4 (33)	3 (27)	10 (27)
Race, *n* (%)				
Asian	1 (7)	3 (25)	1 (9)	5 (14)
Black or African American	2 (14)	2 (17)	3 (27)	7 (19)
More than one race	1 (7)	1 (8)	2 (18)	4 (11)
White	10 (71)	6 (50)	5 (45)	21 (57)
Ethnicity, *n* (%)				
Hispanic or Latino	2 (14)	1 (8)	1 (9)	4 (11)
Not Hispanic or Latino	12 (86)	11 (92)	8 (73)	31 (84)
Prefer not to say	0 (0)	0 (0)	2 (18)	2 (5)
Employment Situation, *n* (%)				
Working (full-time)	9 (64)	2 (17)	5 (45)	16 (43)
Not employed	5 (36)	10 (83)	6 (55)	21 (57)
Education Level, *n* (%)				
High school	0 (0)	0 (0)	2 (18)	2 (5)
Some college	2 (14)	4 (33)	1 (9)	7 (19)
College graduate	6 (43)	4 (33)	6 (55)	16 (43)
Post-graduate degree	6 (43)	4 (33)	2 (18)	12 (33)
Smoking, *n* (%)				
Current smoker	1 (7)	1 (8)	3 (27)	5 (14)
Never smoked	11 (79)	9 (75)	4 (36)	24 (65)
Previously smoked, but quit	2 (14)	2 (17)	4 (36)	8 (22)
BMI Mean (SD)	26.25 (7.00)	21.92 (8.23)	29.38 (8.76)	25.77 (8.29)
Duration of musculoskeletal pain, *n* (%)				
6 months to 1 year	0 (0)	1 (8)	1 (9)	2 (5)
1–5 years	7 (50)	3 (25)	6 (55)	16 (43)
More than 5 years	7 (50)	8 (67)	4 (36)	19 (52)
Number of pain areas or locations related to musculoskeletal pain, *n* (%)				
1	2 (14)	0 (0)	1 (9)	3 (8)
2	2 (14)	3 (25)	0 (0)	5 (14)
3	5 (36)	4 (33)	8 (73)	17 (46)
4	1 (7)	2 (17)	1 (9)	4 (11)
5	2 (14)	1 (8)	0 (0)	3 (8)
More than 5	2 (14)	2 (17)	1 (9)	5 (14)
Pain intensity at location #1, mean (SD)	6.1 (1.99)	5.3 (1.83)	7.2 (1.17)	6.2 (1.84)
Pain intensity at location #2, mean (SD)	6.0 (2.22)	5.4 (1.36)	6.9 (1.58)	6.1 (1.83)
Pain intensity at location #3, mean (SD)	5.4 (2.19)	4.8 (1.49)	5.5 (2.25)	5.3 (1.99)
Primary pain location, *n* (%)				
Back	8 (57)	5 (45)	5 (45)	18 (49)
Neck	2 (14)	1 (9)	3 (27)	6 (16)
Foot	1 (7)	2 (18)	1 (9)	4 (11)
Knee	0 (0)	1 (9)	2 (18)	3 (8)
Hip	3 (21)	0 (0)	0 (0)	3 (8)
Hands	0 (0)	1 (9)	0 (0)	1 (3)
Shoulder	0 (0)	1 (9)	0 (0)	1 (3)
Missing	0 (0)	1 (9)	0 (0)	1 (3)

Note: percentages may not sum to totals because of rounding.

**Table 2 ijerph-19-14875-t002:** Use of pain medication to manage CMP.

	Use of Pain Medication = Yes (*n* (%))		
	In-Person mAPA	Self-Guided mAPA	Control
Baseline	14 (100)	12 (100)	11 (100)
Post-APA	11 (79)	10 (83)	11 (100)
1M follow-up	11 (79)	9 (75)	11 (100)

**Table 3 ijerph-19-14875-t003:** Summary statistics for the study outcomes.

	In-Person mAPA	Self-Guided mAPA	Control	Cohen’s d In-Person vs. Control	Cohen’s d Self-Guided vs. Control	Cohen’s d In-Person vs. Self-Guided	Cronbach’s Alpha
Physical Function (RMDQ)							0.923
Baseline Mean (SD)	5.7 (5.78)	3.8 (4.90)	7.3 (7.82)				
Post-APA Mean (SD) (%) ^†^	3.8 (5.10) (−0.32)	2.9 (4.63) (−0.42)	7.1 (7.27) (−0.06)	0.02	−0.01	0.06	
Mean Change (SD)	−1.1 (3.24)	−1.7 (2.00)	−0.5 (2.25)				
≥30% Reduction, *n* (%) ^†^	4 (29)	3 (25)	0 (0)				
1M Follow-up Mean (SD) (%) ^†^	3.5 (3.50) (−0.29)	2.9 (4.40) (−0.48)	6.0 (7.38) (−0.25)	−0.04	−0.03	−0.02	
Mean Change (SD)	−1.4 (2.50)	−0.1 (2.85)	0.3 (6.66)				
≥30% Reduction, *n* (%) ^†^	4 (29)	3 (25)	0 (0)				
Pain Intensity (Average)							Single item
Baseline Mean (SD)	5.4 (1.69)	5.5 (1.44)	6.8 (2.04)				
Post-APA Mean (SD) (%) ^†^	3.0 (1.41) (−0.45)	2.6 (1.51) (−0.48)	6.5 (1.75) (−0.01)	−1.45	−1.25	0.09	
Mean Change (SD)	−2.5 (1.37)	−2.7 (2.11)	−0.4 (1.63)				
≥30% Reduction, *n* (%) ^†^	7 (50)	6 (50)	0 (0)				
1M Follow-up Mean (SD) (%) ^†^	3.2 (1.40) (−0.42)	2.9 (1.83) (−0.45)	5.7 (1.64) (−0.11)	−0.91	−0.81	0.05	
Mean Change (SD)	−2.4 (1.43)	−2.4 (2.01)	−1.5 (2.12)				
≥30% Reduction, *n* (%) ^†^	8 (57)	5 (42)	0 (0)				
Pain Catastrophizing							0.949
Baseline Mean (SD)	13.4 (14.12)	15.1 (11.86)	22.7 (17.07)				
Post-APA Mean (SD) (%) ^†^	8.6 (9.96) (−0.47)	11.2 (6.89) (−0.25)	16.7 (13.21) (−0.23)	0.09	−0.07	0.09	
≥30% Reduction, *n* (%) ^†^	6 (43)	6 (50)	0 (0)				
1M Follow-up Mean (SD) (%) ^†^	6.0 (5.67) (−0.52)	7.4 (6.54) (−0.47)	17.3 (10.71) (−0.13)	−0.13	−0.02	−0.14	
≥30% Reduction, *n* (%) ^†^	7 (50)	7 (58)	0 (0)				
Fear Avoidance of Physical Activity							0.879
Baseline Mean (SD)	13.1 (8.28)	11.6 (5.54)	14.7 (10.02)				
Post-APA Mean (SD) (%) ^†^	10.3 (7.04) (−0.30)	11.2 (7.08) (0.02)	13.6 (9.76) (−0.14)	0.13	0.48	−0.46	
≥30% Reduction, *n* (%) ^†^	7 (50)	2 (17)	0 (0)				
1M Follow-up Mean (SD) (%) ^†^	9.5 (6.52) (−0.22)	6.6 (6.02) (−0.51)	17.1 (8.25) (0.09)	−0.61	−1.40	0.12	
≥30% Reduction, *n* (%) ^†^	5 (36)	7 (58)	0 (0)				
Anxiety							0.972
Baseline Mean (SD)	51.0 (10.20)	53.4 (11.95)	51.6 (9.23)				
Post-APA Mean (SD) (%) ^†^	51.1 (8.68) (−0.01)	52.8 (11.89) (−0.03)	57.5 (10.64) (0.12)	−1.07	−1.11	0.10	
1M Follow-up Mean (SD) (%) ^†^	50.9 (10.19) (−0.02)	52.8 (9.90) (0.00)	55.2 (9.06) (0.09)	−0.60	−0.55	−0.08	
Depression							0.965
Baseline Mean (SD)	49.8 (7.52)	49.6 (7.84)	48.6 (7.84)				
Post-APA Mean (SD) (%) ^†^	49.6 (8.58) (−0.02)	46.3 (7.08) (−0.07)	53.2 (10.69) (0.10)	−0.95	−1.27	0.49	
1M Follow-up Mean (SD) (%) ^†^	47.9 (8.07) (−0.05)	47.6 (8.50) (−0.03)	50.4 (9.89) (0.04)	−0.92	−0.63	−0.17	
Fatigue							0.986
Baseline Mean (SD)	53.5 (10.07)	54.7 (13.24)	55.2 (11.21)				
Post-APA Mean (SD) (%) ^†^	50.6 (8.21) (−0.07)	53.5 (12.02) (−0.00)	57.7 (13.31) (0.05)	−0.96	−0.53	−0.35	
1M Follow-up Mean (SD) (%) ^†^	47.9 (9.33) (−0.11)	54.4 (17.05) (−0.03)	57.7 (12.76) (0.05)	−1.08	−0.45	−0.45	
Sleep							0.913
Baseline Mean (SD)	53.5 (6.83)	50.3 (7.01)	54.9 (8.01)				
Post-APA Mean (SD) (%) ^†^	49.6 (4.45) (−0.07)	51.8 (4.19) (−0.00)	55.3 (7.35) (0.02)	−0.96	−0.53	−0.35	
1M Follow-up Mean (SD) (%) ^†^	52.3 (4.19) (−0.02)	49.9 (6.25) (−0.04)	56.5 (6.76) (0.04)	−1.08	−0.45	−0.45	

Note: ^†^ indicates percentage change from T1 equal to change/T1; T-score was used for anxiety, depression, fatigue, and sleep; denominator for achieving 30% reduction was the analysis sample; participants lost to follow-up were treated as not achieving 30% reduction.

**Table 4 ijerph-19-14875-t004:** Satisfaction with APA.

	In-Person mAPA *n* (%)	Self-Guided mAPA *n* (%)
One week after APA		
Completely satisfied	1 (7)	3 (25)
Somewhat satisfied	9 (64)	7 (58)
Not satisfied	2 (14)	0 (0)
Missing	2 (14)	2 (17)
Two weeks after APA		
Completely satisfied	2 (14)	3 (25)
Somewhat satisfied	9 (64)	6 (50)
Not satisfied	1 (7)	1 (8)
Missing	2 (14)	2 (17)
Three weeks after APA		
Completely satisfied	2 (14)	5 (42)
Somewhat satisfied	9 (64)	4 (33)
Not satisfied	0 (0)	1 (8)
Missing	3 (21)	2 (17)
Four weeks after APA		
Completely satisfied	1 (7)	3 (25)
Somewhat satisfied	10 (71)	6 (50)
Not satisfied	0 (0)	1 (8)
Missing	3 (21)	2 (17)
One month after 4 weeks of APA (1M follow-up)		
Completely satisfied	1 (7)	5 (42)
Somewhat satisfied	10 (71)	3 (25)
Not satisfied	0 (0)	1 (8)
Missing	3 (21)	3 (25)

## Data Availability

Data are available upon request.

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
