# Peer review of "Self-Guided Smartphone Application to Manage Chronic Musculoskeletal Pain: A Randomized, Controlled Pilot Trial"

_ijerph, 2022, doi:10.3390/ijerph192214875_

Round 1

Reviewer 1 Report

Thank you for the opportunity to review a very interesting research.

Abstract:

The numbers of participants should be given in regular brackets.

Line 25 - Please avoid terms like “many patients”.

General comments:

Throughout the text, I suggest placing references numbers in square brackets before the period ending the sentence.

Introduction:

No comments.

Methods:

No comments.

Results

Why didn't the authors perform a statistical comparison between In-person mAPA group and self-guided mAPA group, but only between each group and the control group?

Discussion:

It would be worthwhile for the discussion to touch on the effectiveness of interventions in the context of pain and functional level?

There is no indication of the limitations of the study.

Reviewer 2 Report

I am pleased to review this manuscript. This paper is written comprehensively, but there are some aspects needed for revision.

 Abstract:

The abstract was described in a structured way. I suggest to add measurement in the method section.

Introduction:

In this part, it is better to emphasize the importance of studying chronic musculoskeletal pain and supplement the current epidemiological situation, such as morbidity and mortality.

Materials and Methods:

1.Theoretical framework: Make the study design and setting section more informative. In discussion part, you said ‘tested our refined mAPA in this study using a theory-driven (Bandura’s self-efficacy) foundation and motivational messages.’ Please add specific theoretical framework and its application in this study.

2.Sample size: This is really a point of concern. The sample size is small in this study. How to calculate the sample size?

3.Instruments: Please add the reliability and validity of scales and questionnaires.

4.Quality control: How to address rigour of this study?

Results:

The results present match the methods described, and provide more advances in the field.

Discussion:

1.This section is a little brief. I suggest that the first paragraph summarizes the findings and innovation of the study, and the subsequent part is elaborated in points.

2.Lack of comparison and in-depth reflection with previous studies. And I suggest introducing more implication for practice regarding to the results of this study in discussion.

3.What is the limitation of this study?
